# Curcumin Administration Routes in Breast Cancer Treatment

**DOI:** 10.3390/ijms252111492

**Published:** 2024-10-26

**Authors:** Bianca Mayo, Silvana Penroz, Keila Torres, Layla Simón

**Affiliations:** Nutrition and Dietetic School, Universidad Finis Terrae, Santiago 7501014, Chile; bmayor1@uft.edu (B.M.); spenrozb@uft.edu (S.P.)

**Keywords:** Curcuma, natural products, mechanisms, pharmacological activity, oral, intravenous

## Abstract

Breast cancer is a public health concern worldwide, characterized by increasing incidence and mortality rates, requiring novel and effective therapeutic strategies. Curcumin is a bioactive compound extracted from turmeric with several pharmacological activities. Curcumin is a multifaceted anticancer agent through mechanisms including the modulation of signaling pathways, inhibition of cell proliferation, induction of apoptosis, and production of reactive oxygen species. However, the poor water solubility and bioavailability of curcumin create important barriers in its clinical application. This review elaborates on the therapeutic potential of curcumin in breast cancer treatment, focusing on the efficacy of different administration routes and synergistic effects with other therapeutic agents. The intravenous administration of curcumin-loaded nanoparticles significantly improves bioavailability and therapeutic outcomes compared to oral routes. Innovative formulations, such as nano-emulsifying drug delivery systems, have shown promise in enhancing oral bioavailability. While intravenous delivery ensures higher bioavailability and direct action on tumor cells, it is more invasive and expensive than oral administration. Advancing research on curcumin in breast cancer treatment is essential for improving therapeutic outcomes and enhancing the quality of life of patients.

## 1. Introduction

Breast cancer represents a significant global health challenge due to its widespread prevalence and the predicted increase in its incidence in the future [1]. Breast cancer is the most common type of cancer among women worldwide, particularly in those over 40 years of age. A 35% increase in the prevalence of breast cancer is estimated by the year 2040. Furthermore, deaths attributed to breast cancer are projected to rise by over 50%, from 685,000 in 2020 to 1 million by 2040 [1]. Breast cancer is classified into different subtypes based on the presence or absence of specific protein markers, namely estrogen receptor α (ERα), progesterone receptor (PR), and ERBB2 (also known as Human Epidermal Growth Factor Receptor 2, HER2). Hormone-dependent subtypes, such as ER+ and PR+ breast cancers, typically have a more favorable prognosis, as they respond well to hormone therapies like tamoxifen, which target the hormone signaling pathways that drive cancer growth. HER2+ breast cancers, on the other hand, respond to monoclonal antibody treatments such as pertuzumab, which binds to an extracellular domain of the HER2 receptor. In contrast, triple-negative breast cancer (TNBC), characterized by the absence of ERα, PR, and ERBB2 expression, is considered more aggressive and presents significant therapeutic challenges, often resulting in poorer clinical outcomes due to the lack of targeted therapies [2,3]. For this reason, identifying novel alternative treatments for breast cancer is essential to reduce patient suffering and death rates.

Curcumin is the primary polyphenolic active compound derived from turmeric of Curcuma longa [4], a phytoextract exhibiting orange-yellow pigmentation. Widely used in Asia, curcumin serves as a food additive and coloring agent [5]. It is virtually insoluble in aqueous solutions at neutral and acidic pH levels. However, due to its lipophilic nature, it can be dissolved in some organic solvents, such as methanol, ethanol, acetone, and dimethyl sulfoxide [6].

Curcumin is associated with a wide range of health benefits. It is commonly used to treat conditions such as menstrual pain, cardiovascular disorders, and cough relief [7]. Its cardiovascular benefits include reducing blood pressure, lowering cholesterol and triglyceride levels, and inhibiting platelet aggregation. Additionally, curcumin inhibits the action of intestinal amylases and GLUT1 glucose transporters in enterocytes, thereby reducing intestinal absorption and blood glucose levels. Curcumin also suppresses the secretion of pro-inflammatory cytokines, including IL-4, IL-6, IL-8, and TNF-α, while stimulating the production of anti-inflammatory cytokines like IL-10 and soluble intercellular adhesion molecule-1 (sCAM-1) [5]. Curcumin exhibits antibacterial, antifungal, and antiviral properties, effectively inhibiting the activity of certain bacteria, such as *Escherichia coli* and *Staphylococcus aureus* [8,9]. Curcumin also combats strains of *Candida*, *Cryptococcus*, *Aspergillus*, *Trichosporon,* and *Paracoccidioides* [10] and inhibits viral replication [11]. Curcumin has antioxidant effects, scavenging free radicals through its phenolic ring structure bearing a methoxy group and stimulating the production of the antioxidant glutathione, thus protecting cells from oxidative damage [12]. In neuroprotection, curcumin facilitates the degradation of β-amyloid in brain tissue, preventing its accumulation and protecting neurons by reducing hyperphosphorylation of tau protein [13,14].

The antitumoral properties of curcumin relate to the regulation of cellular pathways. Curcumin is recognized for its efficacy in cancer therapy by inducing apoptosis, inhibiting cell proliferation, and preventing the transformation of normal into tumoral cells [4,15]. The induction of apoptosis by inhibiting signaling pathways such as NF-κB, PI3K/Akt, and MAPK prevents tumor cell survival [16]. Curcumin increases the production of reactive oxygen species (ROS), leading to oxidative stress and mitochondrial membrane rupture in cancer cells [15,17]. Additionally, it prevents the development and progression of cancer by inhibiting signaling pathways such as COX-2, CD-31, VEGF, and IL-8 and by controlling the expression of pro-tumoral oncogenic proteins [18]. Moreover, curcumin reduces cell growth by inhibiting the production of nitric oxide and endothelial nitric oxide synthases (eNOS). Curcumin regulates cellular metastasis by reducing the levels of matrix metalloproteinases (MMP) such as MMP-9, MMP-2, and VEGF, thereby inhibiting migration, invasion, and angiogenesis [17]. Furthermore, curcumin recruits and activates immune system cells like microglia and natural killer cells to induce apoptosis [5].

In the context of breast cancer, curcumin has been shown to be effective in ER+ and TNBC [19]. In vitro, curcumin reduces proliferation, induces apoptosis, and inhibits the spread of breast cancer cells. In vivo, curcumin reduces tumor growth in experimental breast cancer models [20]. In clinical trials, curcumin prevents breast cancer progression and decreases tumor markers when administered with docetaxel [21]. Moreover, curcumin reduces the side effects of conventional cancer therapy, such as radiation dermatitis [22,23].

In this sense, curcumin has emerged as a promising candidate for breast cancer treatment. In vitro and in vivo studies have shown that curcumin demonstrates antitumoral and antiproliferative properties in the context of breast cancer. Recent research indicates that concurrently using curcumin with other chemotherapeutic agents significantly increases apoptosis within cancer cells [24]. Positive results have been documented with the intravenous administration of curcumin in cases of advanced and metastatic breast cancer [25,26]. Nanoemulsions containing keratin and curcumin demonstrate enhanced absorption by breast cancer cells and exert cytotoxic effects, resulting in a significant reduction in cell viability [27]. The oral administration of curcumin offers challenges due to its low water solubility and reduced bioavailability [7]. Some strategies have been developed to enhance the bioavailability and therapeutic efficacy of curcumin, including combinations with other compounds [28] and encapsulation techniques for its delivery [15]. This review aims to analyze existing literature, focusing on the correlation between breast cancer treatment outcomes and the therapeutic efficacy of curcumin, with a particular emphasis on the influence of different administration routes.

## 2. Mechanisms of Action of Curcumin on Breast Cancer

Breast cancer is a pathology affecting the mammary glandular tissue, where epithelial cells undergo uncontrolled cell division. Tumor formation, known as in situ cancer, can initiate in the ducts or lobules. It may subsequently metastasize through lymphatic and blood vessels, spreading to lymph nodes and distant tissues [29].

Several factors contribute to an increased risk of breast cancer, including advanced age, obesity, physical inactivity, family history or genetic predisposition (BRCA1, BRCA2, PALB2, TP53, PTEN, STK11, NF1), exposure to exogenous hormones or radiation, and alcohol and tobacco consumption [30,31,32].

Curcumin has antitumor effects, particularly in breast cancer treatment [33]. Research indicates that curcumin is involved in several biological processes that synergistically lead to a reduction in tumor volume, underscoring its potential as a multifaceted agent in breast cancer therapy (Figure 1) [33,34,35,36].

Curcumin inhibits the activity of the β-catenin protein, a crucial component of the Wnt/β-catenin signaling pathway, which regulates stem cell differentiation, tissue regeneration, and cell proliferation (Table 1) [37,38]. Furthermore, curcumin blocks the PI3K/Akt pathway, preventing angiogenesis, growth, and cell proliferation, while promoting senescence. In this way, curcumin leads to autophagy and apoptosis of breast cancer cells [39]. Curcumin also reduces the activation of the epidermal growth factor receptor (EGFR) signaling pathway, which is responsible for cell proliferation and tumor growth. By decreasing the amount of EGFR proteins on the cell membrane, curcumin reduces sensitivity to its ligands, thereby inhibiting tumor cell proliferation [4,34].

NF-κB is a key transcription factor that contributes to the proliferation of breast cancer cells, typically activated by TNF-α and various interleukins, and plays a crucial role in oncogenic gene transcription. Curcumin inhibits the phosphorylation of NF-κB, effectively suppressing these transcription processes and blocking proliferative pathways. Additionally, curcumin induces the production of ROS and modulates the Bax-mediated apoptotic pathway in breast cancer [25,40,41,42,43].

Curcumin regulates the microRNA (miRNA) and messenger RNA (mRNA) interaction network, affecting cell cycle progression, migration, and invasion in a cell type-specific manner within breast cells [44]. Furthermore, curcumin induces the upregulation of tumor-suppressive miR-181b, miR-34a, miR-16, miR-15a, and miR-146b-5p while downregulating the expression of oncogenic miR-19a and miR-19b in breast cancer cells. The regulatory impact on miRNA expression significantly contributes to the suppression of tumorigenesis and the induction of apoptosis [45].

Curcumin blocks the binding of cyclin D1 to Cyclin-dependent kinase 4 (CDK4), consequently diminishing its activity and halting cellular division. Additionally, curcumin promotes the degradation of cyclin E, a protein often overexpressed in breast cancer, while simultaneously activating Cyclin-dependent kinase (CDK) inhibitors. These effects arrest cell division during the G1 phase. Moreover, curcumin induces DNA degradation and apoptosis in breast cancer cells [46,47,48]. In TNBC, curcumin inhibits proliferation, invasion, and migration through overexpression of Glioma-associated oncogene homolog-1 (Gli1) and regulation of the Hedgehog (Hh)/Gli1 pathway [49]. These mechanisms highlight curcumin as a potential therapeutic agent in modulating tumor growth and underscore its significance in developing cancer treatment strategies.

**Table 1 ijms-25-11492-t001:** Summary of formulations, models, doses and mechanisms of curcumin use in breast cancer.

Formulation	Model	Doses	Mechanism	Reference
Curcumin in corn oil	BALB/c mice with 4T1-luc cells	100 mg/kg through gavage	Inhibits the Wnt/β-catenin signaling pathway, which regulates stem cell differentiation, tissue regeneration, and cell proliferation	[38]
Curcumin in dimethyl sulfoxide (DMSO)	MCF-7 cells	30 µM for 24 h	Blocks the PI3K/Akt pathway, preventing angiogenesis, growth, and cell proliferation while promoting senescence, autophagy and apoptosis	[39]
Curcumin-loaded PLGA-PEG nanoparticles	MCF-7 cells BALB/c mice with MCF-7 cells	10 µg/mL for 2 h 5 mg/kg in mice	Decreases the amount of EGFR proteins on the cell membrane, inhibiting sensitivity to its ligands, thereby preventing tumor cell proliferation	[34]
Hyaluronic acid-tagged mesoporous silica nanoparticles loaded with curcumin (MSN-HA-C)	MDA-MB-231 cells Swiss albino mice with EAC cells	12 μg/mL for 48 h 67.75 mg/kg in mice	Causes cell death by the induction of NF-κB and Bax-mediated pathway of apoptosis Inhibits cell migration	[42]
Free curcumin	MCF-7 cells	1.25 mg/mL for 24 h	Downregulates the Raf-1 oncogene and suppresses telomerase activity, alongside upregulating the TNF-α and IL-8 cytokines	[43]
Apoferritin nanoparticles loaded with Quercetin and Curcumin (Que-Cur-HoS-Apo NPs)	MCF-7 and MCF-10A cells	2.74 μM for 48 h	Promotes cytotoxicity, apoptosis, ROS production	[50]
Folic acid-modified curcumin-loaded liposomes (LIP-CCM-FA)	Two-dimensional (2D) and three-dimensional (3D) cell culture models	50 µM for 24 h	Promotes cytotoxicity and enhancing cellular and spheroid penetration	[51]
Chitosan-coated liposomes encapsulating curcumin	MCF-7 cells	12.5 μM for 96 h	Inhibits growth	[52]
Curcumin-Loaded Solid Lipid Nanoparticles (Cur-SLNs)	SKBR3 cells	20 μM for 0, 8, 16 or 48 h	Induces apoptosis, promoting the ratio of Bax/Bcl-2, and decreasing the expression of cyclin D1 and CDK4	[53]

## 3. Strategies for Effective Curcumin Delivery to Breast Cancer Cells

Despite its promising effects in cancer treatment, curcumin faces several challenges that hinder its clinical application. These challenges include its hydrophobic nature, photo-degradability, poor absorption, rapid metabolism and elimination, short half-life, and low bioavailability [4,54]. It is important to note that, when taken orally, curcumin is rapidly metabolized in the liver and intestines through glucuronidation and sulfation at the phenolic site, significantly limiting its absorption and distribution [54]. To address these limitations, various delivery methods aimed at improving curcumin’s pharmacokinetics, including specialized formulations such as nanoparticles, liposomes, phospholipid complexes, as well as strategic drug combinations and alternative routes of administration, have shown promising results [55].

Numerous studies have employed MCF-7 cells to explore the impacts of different curcumin delivery methods on breast cancer. The MCF-7 cell line is ERα+ and PR+, non-invasive, and non-tumorigenic in vivo unless supplemented with estrogen [56]. Khazaei Koohpar et al. (2015) demonstrated that curcumin induces cellular apoptosis, accompanied by a decrease in MCF-7 cell viability [57]. Fawzy et al. (2024) reported that treatment with curcumin led to the downregulation of the Raf-1 oncogene and suppression of telomerase activity, alongside the upregulation of the TNF-α and IL-8 cytokines. These findings suggest that curcumin can be effectively integrated as a component of phytocompound-based therapeutics, particularly focusing on cytokine modulation. Additionally, treatment with curcumin in MCF-7 cells was associated with cellular damage, evidenced by increased lactate dehydrogenase release, indicative of necrosis [43].

The effectiveness of curcumin encapsulated in nano-micelles has been evaluated, demonstrating a decrease in the viability of MCF-7 cells [58]. Nanoparticles loaded with curcumin and quercetin significantly enhanced absorption and cytotoxicity in these cells [50]. Moreover, when curcumin and quercetin were encapsulated in nanocochleates, a synergistic effect was observed in inhibiting MCF-7 cancer cells, suggesting that this is a suitable delivery option for adjuvant therapy in breast cancer [59]. A nanocomposite containing polyacrylic acid, starch, titanium dioxide and curcumin demonstrated enhanced effects in the induction of apoptosis by increasing the bioavailability and controlling the release of the drug compared to free curcumin. Using this nanocomposite for targeted drug delivery minimizes side effects and enhances the effectiveness of the treatment [60,61].

In the cancer microenvironment, the release of curcumin from the curdlan-curcumin complex induces a significant increase in cellular mortality, approximately twice that of unbound curcumin. This heightened efficacy correlates with enhanced apoptosis in MCF-7 cells, as evidenced by a notable increase in the Bcl-2/Bax ratio [62]. Biocompatible metal–organic frameworks (bioMOFs) loaded with curcumin and encapsulated with folate exhibit higher cytotoxicity. This is characterized by reduced cell viability and increased rates of apoptosis and necrosis in MCF-7 and 4T1 cell lines. These findings highlight the potential of folate to enhance the specificity and efficacy of bioMOF materials for curcumin delivery by leveraging the overexpression of folate receptors on cancer cells [63]. Similar results were obtained with folic acid-modified curcumin-loaded liposomes, demonstrating greater cytotoxicity, enhanced cellular internalization, and improved spheroid penetration compared to free curcumin [51].

Omrani et al. (2023) developed a pH-sensitive nanocarrier composed of chitosan, starch, and a MoS_2_ nanocomposite. This formulation enhances the capacity to load curcumin, induces late-stage apoptosis, and reduces the viability of MCF-7 cells [61]. Similarly, another pH-sensitive formulation consisting of Fe_2_O_3_, chitosan, and Carbon Quantum Dots (CQDs) led to a notable reduction in MCF-7 cell viability [64].

Chitosan-coated liposomes encapsulating curcumin demonstrated an 88% loading efficiency compared to 65% for non-coated liposomes, significantly enhancing the growth-inhibitory effect on MCF-7 cells [52]. The development of a dual pH/redox-responsive nanocarrier system, which incorporates hyaluronic acid-decorated hollow meso-organosilica/poly(methacrylic acid) nanospheres, has been specifically engineered to deliver curcumin to breast cancer cells. The release dynamics of curcumin from these nanoparticles are influenced by environmental factors, particularly the acidic pH characteristic of tumor tissues, which improves the bioavailability of the treatment. These nanoparticles demonstrate elevated cytotoxicity against the MCF-7 breast cancer cell line, attributed to enhanced internalization facilitated by CD44 receptors [65].

In vitro studies conducted on MDA-MB 231 cells, a TNBC cell line characterized by elevated expression levels of genes linked to epithelial–mesenchymal transition (EMT), migration, and cellular differentiation [56], showed the effectiveness of a hydrogel formulation containing laponite rapid dispersion (Lap^®^), chitosan and polyvinyl alcohol, designed specifically for the controlled delivery of curcumin (Lap^®^CUR/CS@PVA hydrogel). The cells treated with this hydrogel exhibited nuclear fragmentation and chromatin condensation, indicative of cell death resulting from curcumin release [66]. Similar results were observed when curcumin was loaded onto solid lipid nanoparticles (SLN) in SKBR3 cells that overexpress HER2, resulting in increased apoptosis [53].

In summary, several delivery methods, such as nano-micelles, nanocomposites, and pH-sensitive nanocarriers, have been developed to enhance the effectiveness of curcumin. Promising results in studies on breast cancer cells indicate that curcumin can increase cytotoxicity and induce apoptosis.

## 4. The Synergy of Curcumin and Breast Cancer Treatments in Cell Line Studies

The synergistic effects of curcumin with various therapies for treating breast cancer have been extensively documented. For instance, combining curcumin with paclitaxel inhibits the NF-κB signaling pathway and induces apoptosis in breast cancer cells to a greater extent compared to the administration of either compound alone [40]. Additionally, the combination of curcumin with paclitaxel reduces the expression of c-Ha-Ras, Rho-A, and p53 genes [67]. Alemi et al. (2018) administered paclitaxel and curcumin in PEGylated niosomal (non-ionic surfactant vesicles) formulations to MCF-7 cells, enhancing cytotoxic activity and inhibiting cell growth [68]. Similarly, nanoparticles loaded with paclitaxel and curcumin effectively suppress 4T1 and MDA-MB-231 cell viability, while exhibiting no adverse effects on the survival of non-cancerous cells. Furthermore, curcumin inhibits the production of P-glycoprotein and reverses multidrug resistance in breast cancer cells [69].

Curcumin also reverses resistance to doxorubicin in breast cancer cells (MCF-7 and MDA-MB-231) by targeting the ATPase activity of ABCB4 [70]. Recently, Moghadam and colleagues (2024) designed a biosystem featuring a distinctive L-lysine amino acid coating on the surface of magnetic graphene oxide@rod-Cu(II) metal–organic frameworks (GO-Fe_3_O_4_@Cu MOFs-Lys). The L-lysine amino acid within this biocompatible structure enables efficient drug loading and precise/targeted release within breast cancer cells. MCF-7 cells show decreased viability when doxorubicin and curcumin are loaded into the biosystem [71]. Saharkhiz et al. (2023) revealed that the co-loading of doxorubicin and curcumin within a pH-responsive niosomal formulation facilitates a synergistic interaction by amplifying their cytotoxic effects in MCF-7 cells [72]. Similarly, a metal–organic framework (MOF) coated with folic acid-activated chitosan (FC) promotes the concurrent delivery of doxorubicin and curcumin, enhancing drug accumulation at the target site facilitated by folate receptor interaction. Moreover, the cytotoxicity profile of the curcumin and doxorubicin-loaded FC-MOF is superior to that of the free drugs in MCF-7 breast cancer cells, suggesting a greater therapeutic efficacy with reduced adverse effects [73].

In TNBC cells, curcumin enhances the inhibitory effects on cell proliferation, reverses morphological changes, and prevents the doxorubicin-induced downregulation of E-cadherin expression. Curcumin inhibits the doxorubicin-induced epithelial–mesenchymal transition (EMT) by targeting the TGF-β and PI3K/AKT signaling pathways [17]. Wang et al. (2022) reported that curcumin sensitizes TNBC to the anticancer effects of carboplatin by upregulating ROS production, leading to the downregulation of the DNA repair protein RAD51, thereby inhibiting proliferation and inducing apoptosis [74].

Moreover, curcumin provides significant protection against 5-Fluorouracil (5-FU) cytotoxicity by a factor of 7–10 times, acting as an adjuvant therapy and allowing for higher doses or longer treatment durations by shielding normal cells from reduced viability [75]. Curcumin enhances the sensitivity of breast cancer cells to 5-FU-induced apoptosis through a mechanism that involves the downregulation of NF-κB, mediated by thymidylate synthase (TS) [76].

The formulation of tamoxifen–curcumin (TMX–Cur)-loaded niosomes induces the upregulation of Bax and p53 gene expression, leading to apoptosis in MCF-7 cells, as well as an increase in the percentage of cells in the Sub-G1 phase compared to treatment with each agent alone [77]. Furthermore, micelles encapsulating curcumin with methotrexate, an anti-metabolite that disrupts DNA, RNA, thymidylate, and protein production, lead to lower survival rates in MCF-7 cells, indicating a greater antitumor effect [78].

Mahmoudi et al. (2021) intercalated curcumin (CUR) into a double-layered membrane of cisplatin (Cis) nanoliposomes (NLP), resulting in a dosage-controlled co-delivery formulation named CUR-Cis@NLP. This formulation demonstrated notable anticancer effects, reducing breast cancer cell viability by 82.5% at optimized concentrations, with induction of apoptosis [79]. Additionally, a delivery system comprising a zein–laponite coacervate was effectively formulated for targeted therapy against MDA-MB-231 cells, enabling the simultaneous administration of cisplatin and curcumin and reducing the cisplatin dosage [80].

These findings underscore the potential of curcumin to improve the efficacy of traditional breast cancer therapies, to safeguard normal cells against side effects, and to enhance the susceptibility of cancer cells to treatments, collectively suggesting improved clinical outcomes.

## 5. Curcumin in Breast Cancer Treatment on Animal Models and Clinical Trials

Curcumin has been demonstrated to offer an effective and safe therapeutic option for breast cancer. The efficacy of curcumin formulations has been established in cell line studies, in vivo models, and clinical trials [21,25,81,82]. Moreover, multiple administration routes for these curcumin-based treatments have been examined, including intratumoral, transdermal, intraperitoneal, oral, and intravenous methods (Figure 2).

### 5.1. Intratumoral Route

The direct delivery of treatments into tumors has become a promising approach, enabling high concentrations of therapeutic agents locally and yielding potent anti-tumor effects. The effectiveness of many local therapies relies on adequate tumor exposure, influenced by the pharmacokinetic properties of the agent [83]. Mahalunkar et al. (2019) demonstrated the antitumor efficacy of folate–curcumin-loaded gold–polyvinylpyrrolidone nanoparticles (FA–CurAu–PVP NPs) in 6-week-old female BALB/c mice with 4T1 tumors that were treated intratumorally. The improved effectiveness of curcumin in this nanoformulation arises from its successful targeting of tumors via folate receptors, coupled with the sustained, gradual release of the drug directly at the tumor site [84]. Similar results were reported in mice bearing TNBC cells (BT-549 cells) that received intratumoral injections of curcumin-loaded phosphorylated calixarene micelles, resulting in inhibited tumor growth [85]. Li et al. (2020) studied the effect of thiolated chitosan-coated liposomal hydrogels as curcumin carriers (CSSH/Cur-Lip gel) injected into tumor-bearing mice (4T1 cells) after tumor resection. The CSSH/Cur-Lip gel exhibited the lowest in situ tumor recurrence rate and the longest survival, with significantly reduced toxicity. This gel provides an extracellular matrix extending curcumin release and inhibiting tumor recurrence [86].

### 5.2. Transdermal Route

The administration of curcumin via the transdermal route as a treatment for breast cancer exhibits a viable alternative. Atlan and Neman (2019) used a transdermal hydrogel loaded with curcumin, which can be absorbed through the dermis and has a chemo-preventive effect [87]. In this regard, ultra-deformable nanovesicles loaded with curcumin have been tested ex vivo on the skin of male albino mice, showing good transdermal penetration [88]. Additionally, tetrahydrocurcumin (THC), a major metabolite of curcumin, encapsulated within chitosan (Ch)-coated nanostructured lipid carriers (NLCs) (THC-Ch-NLCs), exhibited enhanced permeability in human skin assays. This augmented permeation is attributed to a positively charged polymer, such as chitosan, which plausibly acts as a permeation enhancer through interactions with the negatively charged biological membrane. Furthermore, cytotoxicity assays suggest that THC-Ch-NLCs hold considerable promise for the therapeutic intervention of invasive triple-negative breast cancer in vitro [89].

### 5.3. Intraperitoneal Route

The intraperitoneal route, the most invasive method of drug delivery, involves injecting drugs directly into the abdominal cavity. This route allows the administration of larger volumes without overburdening the cardiovascular system and provides the advantage of prolonged circulation time with lower liver uptake compared to intravenous injection. It is commonly used in various in vivo disease model studies [90]. Curcumin reduced tumor growth at concentrations of 50 µg/kg and 200 µg/kg when administered intraperitoneally to mice with xenograft MDA-MB-231 breast cancer cells for four weeks [91]. BALB/c mice bearing MDA-MB-468 tumors were treated with intraperitoneal injections of 10 mg/kg curcumin-loaded phospholipid nanoparticles conjugated with epidermal growth factor (EGF) (EGF-Cur-NP) three times per week. EGF-Cur-NP cause a 59.1% retardation of tumor growth compared to empty nanoparticles after eight injections [92]. In addition, Ferreira et al. (2015) studied the effect of 300 mg/kg of curcumin administered intraperitoneally in mice with induced breast cancer. Tumor size was examined using single-photon emission computed tomography, demonstrating that curcumin is effective in reducing tumor size and halting cancer cell proliferation [93]. Furthermore, the combination of curcumin (100 μg/kg) and cisplatin (2 mg/kg), administered via intraperitoneal injections every day for 2 weeks in a nude mouse MCF-7 xenograft model, significantly inhibits tumor growth [94].

Shao et al. (2021) synthesized an anticancer agent by integrating naphthalimide into the structure of methylene blue (MCLOP) and conjugating it with biotinylated curcumin (Cur-Bio), which was used as a chemosensitizer. The efficacy of these compounds was tested in an MCF-7 xenograft mouse model by intraperitoneal injection. MCLOP administered with Cur-Bio was subjected to visible light irradiation (450 nm for 3 min at 2 W/cm^2^), leading to significant tumor suppression and induction of autophagic cell death. A synergistic strategy employing Cur-Bio as a chemosensitizer activated by light achieved significant antitumor effects [95].

### 5.4. Oral Route

The oral route is preferred for administering drugs and nanoparticles primarily due to its advantages, such as ease of ingestion, enhanced patient compliance, and pain avoidance [81]. In this context, curcumin was incorporated into a self-nano-emulsifying drug delivery system (CPCSNEDDS) for oral administration at a dose equivalent to 100 mg/kg targeting 4T1/BALB/c tumors. Treatment with CPCSNEDDS and a curcumin suspension led to reductions in primary tumor growth by 58.9% and 29.5%, respectively, compared to the control group, which received an aqueous gum acacia suspension [96].

An innovative study by Guao et al. (2024) investigated the therapeutic effects of exercise (30 min of swimming) combined with doses of 200 μL of curcumin solution dissolved in corn oil (100 mg/kg) on breast cancer in BALB/c mice. This combination significantly reduced tumor growth by modifying key signaling pathways, including the calcium, Wnt and IL-17 pathways, and the expression of genes such as CAMK1G, WNT5A, and IL-6. Additionally, metabolomic analysis revealed significant impacts on amino and nucleotide metabolism pathways [38].

Additionally, bilosomes coated with D-alpha-tocopheryl polyethylene glycol succinate (TPGS) and loaded with curcumin (TPGS-CUR-Bil) have been developed, exhibiting desirable colloidal stability. Upon oral administration, these bilosomes showed increased penetration through the lamina propria of rat duodenal tissue in ex vivo studies. The TPGS enhances the cytotoxic effects of curcumin against doxorubicin-resistant breast cancer cell lines, underscoring its potential therapeutic impact [97]. Combining nano-micelles made of PLGA (Poly(Lactide-co-Glycolide)) and levan loaded with curcumin exhibits increased curcumin bioavailability compared to formulations with curcumin-loaded PLGA alone in mice with induced breast cancer. Furthermore, the co-administration of curcumin with gemcitabine in PLGA and levan resulted in the inhibition of the NF-κB pathway, thereby enhancing the anticancer effects of both compounds [98].

Self-microemulsifying Drug Delivery Systems (SMEDDS), such as CUR/IR780@SMEDDS, which combines curcumin with IR780, a photo-therapeutic agent, have demonstrated improved oral absorption, bioavailability, and accumulation of curcumin within the tumor. Additionally, enhanced cellular uptake, increased cytotoxicity and inhibition of cell migration and invasion have been observed when combined with near-infrared laser irradiation. These findings suggest the therapeutic potential of curcumin administered through these microemulsions in treating metastatic breast cancer [99].

Attia et al. (2020) studied the effects of a 14-day treatment regimen comprising paclitaxel, curcumin, and vitamin D3 on Ehrlich ascites carcinoma-bearing mice. Remarkably, this combination therapy significantly reduced tumor size, achieving decreases of 51.27% and 43.83% in groups receiving paclitaxel plus vitamin D3 and paclitaxel plus curcumin, respectively. By day 21, the combination therapy maintained its efficacy, demonstrating reductions in tumor volumes of 31.01% and 33.83% relative to the paclitaxel plus vitamin D3 and paclitaxel plus curcumin groups, respectively. The triple therapy exhibited the lowest Aldehyde Dehydrogenase-1 (ALDH-1) expression, a marker for identifying cancer stem cells, compared with paclitaxel alone. These findings highlight the potential of the triple therapy approach in overcoming paclitaxel resistance [100].

Mice orally treated with a nano-suspension loaded with curcumin and docetaxel (10 mg/kg) exhibit a reduction in tumor size. This effect is attributed to the potential inhibitory activity of P-glycoprotein by curcumin, which complements the anticancer effects of docetaxel [101]. In phase I clinical trials, patients with advanced or metastatic breast cancer underwent treatment combining docetaxel and curcumin. Docetaxel was administered intravenously at 100 mg/m^2^ every three weeks on the first day of each cycle, for a total of six cycles. Curcumin was administered orally from 500 mg/day and gradually increased for seven consecutive days per cycle, starting four days before and continuing two days after the docetaxel infusion. This study concluded that the established recommended dose for curcumin is 6 g/day for seven consecutive days every three weeks, administered alongside a standard dose of docetaxel [21] (see Table 2).

In a randomized, double-blind, placebo-controlled clinical trial involving 30 breast cancer patients undergoing radiotherapy, daily oral administration of 6 g of Curcumin (Curcumin C3 Complex^®^) reduced the severity and incidence of moist desquamation of radiation dermatitis, compared to placebo. These findings suggest that curcumin may offer a promising adjuvant therapy to mitigate radiation-induced skin toxicity in breast cancer patients undergoing radiotherapy [23]. A subsequent clinical trial involved 37 breast cancer patients; 26 were assigned to the polyphenol group, which received three capsules per day containing curcumin and other polyphenols from diagnosis to surgery (5 + 2 days), while 11 were in the control group. Additionally, the polyphenol group consumed two capsules between 2 and 6 h before surgery. The study detected the presence of metabolites derived from curcuminoids in the mammary tissues of these patients, suggesting that these metabolites might exert anticancer effects with long-term exposure [82] (see Table 2).

### 5.5. Intravenous Route

Intravenous administration enables immediate drug distribution within the bloodstream [81]. In a study assessing its effectiveness in curcumin and breast cancer research, polyethylene glycol (PEG)–albumin–curcumin nanoparticles were administered intravenously to rats at a concentration of 10 mg/mL. This formulation not only improved the solubility and permeability of curcumin but also exhibited a prolonged release profile of 35 days, surpassing nanoparticles containing only curcumin and albumin [102]. Similarly, nude mice bearing MCF-7 xenografts were treated with injections of curcumin-loaded nanoparticles (Cur-P-NPs), formulated using an amphiphilic block copolymer (MePEG-peptide-PET-PCL) at a concentration of 0.2 mg/kg. The Cur-P-NPs lead to a targeted accumulation of curcumin in the tumors, demonstrating high specificity and enhanced bioavailability [103].

Employing monoclonal graphene oxide nanoparticles (GO NPs) loaded with curcumin, folic acid antibody (FA), single layers of carboxymethylcellulose (CMC), and polyvinylpyrrolidone (PVP, Cur-FA-CMC/PVP GO NPs) demonstrated substantial antitumor activity against 4T1 breast cancer in in vivo mice models. This formulation induced necrosis, inhibited tumor growth, and reduced angiogenesis. The Cur-FA-CMC/PVP GO NPs were designed to target cancer cells by binding to overexpressed folate receptors, significantly enhancing the direct delivery of curcumin to tumor sites and resulting in prolonged survival of the treated mice compared to those receiving control nanoparticles or placebo [104]. In an advanced breast tumor xenograft model using MDA-MB-231 cells in nude mice, curcumin-loaded redox-responsive mesoporous silica nanoparticles (MSN/CUR-PEI-FA) demonstrated significantly enhanced antitumor efficacy. The tumors in mice treated with MSN/CUR-PEI-FA were notably smaller compared to those in control mice and those treated with free curcumin. This suggests that MSN/CUR-PEI-FA can specifically target tumor sites, thereby improving drug delivery and therapeutic outcomes [105].

In 2019, Kundu et al. developed a formulation containing ZnO-PBA-curcumin (zinc oxide, phenylboronic acid, and curcumin), which was administered intravenously at a concentration of 10 mg/kg body weight to tumor-bearing mice. This formulation enhanced the cellular absorption of curcumin compared to free curcumin, thereby improving its efficacy. Additionally, it led to a decrease in tumor mass and volume, an interruption of cell proliferation, and the prevention of metastasis [106]. Jin et al. (2017) utilized PEGylated polylactic-co-glycolic acid nanoparticles loaded with curcumin and GE11, a peptide capable of binding to EGFR. This formulation demonstrated prolonged drug release, inhibition of the EGFR cellular signaling pathway, and consequently, a reduction in tumor size [34].

Li et al. (2023) developed a pH/redox-responsive nanocarrier for curcumin delivery, CUR@PCPP NPs, evaluating its antitumor efficacy in mice with 4T1 tumors. Administered intravenously at a dose of 2 mg/kg curcumin, CUR@PCPP NPs exhibited the strongest antitumor effects, resulting in a low Ki67 signal, indicative of reduced cell proliferation [49]. Karabasz et al. (2019) synthesized a blood-compatible alginate–curcumin conjugate (AA-Cur), facilitating the stable formation of micelles. This formulation demonstrated moderate anti-tumor efficacy against 4T1 cells, with mice treated with AA-Cur exhibiting average tumor masses over 30% lower than those observed in control mice [107].

Moreover, the antitumor efficacy of curcumin nanocrystals modified with hyaluronic acid (HA@Cur-NC) was assessed in female BALB/c mice with 4T1 tumors, showing enhanced anticancer effects through sustained increased tumor permeability [108]. Similarly, mifepristone and curcumin encapsulated in hyaluronic acid-modified liposomes (CUR&RU486/HA-LIPs) significantly reduced tumor volume and growth in a 4T1 + NIH/3T3-bearing mice model. This reduction was attributed to enhanced permeability and retention effects, HA-CD44 receptor-mediated internalization, and decreased extracellular matrix deposition and angiogenesis [109].

Curcumin-loaded, hyaluronic acid-coated zeolitic imidazolate framework-8 nanoparticles (Cur@ZIF-8@HA), administered at a dosage of 25 mg/kg to BALB/c mice with 4T1 cell tumors, significantly reduced tumor sizes. They exhibited a tumor inhibition rate of 77.8%, superior to the Cur@ZIF-8 group (25.9%) and saline controls. Histological analysis indicated increased necrosis and apoptosis in the Cur@ZIF-8@HA-treated tumors, along with a notable reduction in metastasis [110]. Additionally, curcumin conjugates to hyaluronic acid on mesoporous silica nanoparticles (MSN-HA-C) accumulates in the tumor owing to the enhanced permeability and retention effect and HA receptor interaction, despite renal clearance and heart and lung deposition. In vivo studies in Swiss albino mice with EAC solid tumors demonstrated that MSN-HA-C considerably decreased tumor volume and mass, confirming its effectiveness as a curcumin delivery system in cancer therapy [42]. Furthermore, curcumin-loaded oxidation-responsive mPEG-b-PLG (Se)-TP polymeric micelles (Cur-loaded micelles) have prolonged circulation in the bloodstream, efficient tumor accumulation, and drug release in BALB/c mice with 4T1 breast cancer tumors. This formulation induces apoptosis in cancer cells and inhibits tumor angiogenesis and proliferation [111].

Intravenous administration of 50 mg/kg of 99mTechnetium (99mTc) radiolabeled hyaluronan–cholesteryl hemisuccinate conjugates (HA-CHEMS) and D-α-tocopheryl polyethylene glycol succinate (TPGS) in curcumin-loaded nanoparticles (HA-CHEMS-Cur-TPGS NPs) has an antitumor effect in mice bearing 4T1 tumors. This formulation increased tumor cell necrosis and apoptosis, improving mice survival rates [112]. In a related study, 99mTc-labeled curcumin (20 mg/kg) and gemcitabine (16 mg/kg)-loaded nanoparticles (CGNPs), as well as their folate-conjugated counterparts (FCGNPs), were administered to female nude mice with MDA-MB-231 xenografts. Both formulations predominantly accumulated in the liver (approximately 32%) and were excreted via the kidneys (15–18%) four hours post-injection. The FCGNPs exhibited superior tumor targeting and efficacy, achieving up to 94% tumor growth inhibition, thus highlighting the enhanced effectiveness of folate-conjugated nanoparticles in tumor suppression [113].

N-methyl nitroso urea (MNU), a direct-acting carcinogen, was used to induce mammary tumors in 21-day-old female Sprague Dawley rats, chosen for its simplicity and effectiveness in tumorigenesis. Four months after induction, when palpable mammary tumors were present in approximately 80% of the animals, a treatment regimen was initiated involving intravenous injections of methotrexate and curcumin co-encapsulated in poly lactic-co-glycolic acid nanoparticles (MTX-CUR-NPs). The treatment was administered once weekly for four weeks, reducing the tumor size [114]. Additionally, a nanogel formulation comprising heparin-poloxamer P403 (HP403) was engineered for the simultaneous loading of curcumin (Cur) and cisplatin hydrate (CisOH) (HP403@CisOH@Cur). This nanogel formulation resulted in a 90% reduction in tumor volume after a 13-day treatment, with no mortality recorded among the treated mice. These findings underscore the considerable efficacy of curcumin and cisplatin co-delivery in suppressing cancer cell proliferation, while also potentially ameliorating adverse effects [115].

Curcumin-loaded biocompatible solid lipid nanoparticles (Cur-SLNs) have demonstrated significant efficacy and safety in preclinical models targeting drug-resistant JC tumors. Notably, therapeutic regimens combining Cur-SLNs with doxorubicin are particularly effective in reducing tumor growth and mass. These combinations achieve notable therapeutic outcomes without inducing systemic toxicity [116]. Co-delivery of paclitaxel and curcumin via biodegradable polymeric nanoparticles (PTX-CUR-NPs) has been validated in an MCF-7 xenograft mouse model. The administration of PTX-CUR-NPs inhibited tumor growth and induced minimal changes in body weight. Notably, PTX-CUR-NPs extended the survival of tumor-bearing mice to 44 days, compared to 32 days for the group receiving free drugs (paclitaxel + curcumin). Immunohistopathological analysis revealed a lower proliferation index, suggesting potential clinical applications for this drug delivery system due to its enhanced therapeutic efficacy and reduced toxicity [117]. Additionally, PEG-PLGA encapsulated paclitaxel and curcumin nanoparticles (PC-NPs) inhibit tumor volume and demonstrate adequate biocompatibility without systemic toxicity, providing an effective antitumor effect in mice bearing MDA-MB-231 tumors [69].

In a clinical trial, a group of women was administered a combination of 300 mg of curcumin and paclitaxel via the intravenous route. Upon evaluation of adverse effects, intravenous curcumin exhibited no significant safety concerns nor compromised quality of life of patients. Additionally, the adjunct of curcumin demonstrated a modest yet discernible effect in ameliorating gastrointestinal symptoms and fatigue. Analysis conducted three months post-treatment revealed a notably higher overall response rate in the curcumin group compared to the placebo cohort. This indicates a sustained and superior therapeutic effect of curcumin, extending beyond the cessation of treatment [25] (see Table 2).

### 5.6. Comparing the Oral and Intravenous Routes of Curcumin Administration

When comparing the administration of curcumin for breast cancer treatment via oral and intravenous routes, research has demonstrated that the oral route is more cost-effective, accessible, and less invasive than its intravenous counterpart (see Table 3). Furthermore, when administered orally, curcumin treatment for breast cancer has shown greater adherence due to better tolerance and the ability to be used for prolonged periods [54]. Despite the numerous advantages of oral administration, the well-known issue of low bioavailability persists when curcumin is administered this way [118]. In contrast, the intravenous route bypasses gastrointestinal digestion and acts directly on cancer cells. In this way, the intravenous route requires lower administration doses and exhibits higher bioavailability than the oral route. However, it is more invasive and expensive, which makes it tedious and less preferable for most individuals undergoing oncologic treatments [54]. For this reason, new methods have been formulated to encapsulate curcumin, thus increasing its bioavailability and, consequently, its efficacy.

## 6. Challenges and Future Perspectives

The therapeutic challenges of using curcumin in breast cancer treatment are primarily related to its low bioavailability, rapid degradation, and poor aqueous solubility [54,55]. Although innovative formulations such as nano-micelles, solid lipid nanoparticles, and folic acid-modified liposomes have been developed to enhance its absorption and efficacy [53,69,96,97,98,99,101,102,103,104,105,106,107,108,109,110,111,112,113,114,115,116,117], translating these strategies into clinical applications remains a significant challenge. Administering curcumin in a way that maintains its potency while reducing side effects (gastrointestinal disturbances such as nausea, diarrhea, bloating, dermatitis, and urticaria) [119] requires further investigation. Additionally, the variability in patient responses and the need for precise control over curcumin release within the tumor microenvironment are also major challenges.

Looking ahead, overcoming these limitations will require the development of more sophisticated drug delivery systems and personalized therapeutic approaches. Combining curcumin with conventional cancer therapies, such as paclitaxel or doxorubicin, has demonstrated enhanced anticancer effects in preclinical studies, highlighting its potential for synergistic treatment strategies [21,25]. Further research is also needed to refine delivery methods, which could improve the ability of curcumin to target tumor cells while minimizing off-target effects [119]. With continued advancements, curcumin could play a more significant role in integrative breast cancer treatment regimens.

Finally, it is essential to note that, according to the U.S. National Cancer Institute, while curcumin has been investigated for its potential role in cancer treatment, findings from early-stage trials require confirmation in larger, well-powered studies assessing its safety and effectiveness. The NIH emphasizes that, at present, there is insufficient evidence to recommend curcumin-containing products for cancer prevention or treatment [120]. Further research is necessary to conclusively establish its safety and therapeutic efficacy.

## 7. Conclusions

Curcumin has demonstrated heterogeneous mechanisms of action, including antiproliferative effects, modulation of various signaling pathways, regulation of gene transcription, and the ability to disrupt tumor cell division. Despite these promising anticancer properties, the challenge of the bioavailability of curcumin remains a significant barrier to its therapeutic efficacy. Innovative pharmacological formulations have been developed to enhance the bioavailability of curcumin through various administration routes, with oral and intravenous routes being the most prevalent. While the oral route offers advantages in non-invasiveness, cost-effectiveness, and tolerance, it suffers from lower bioavailability than the intravenous route. This underscores the ongoing need for research to optimize oral delivery methods and explore more efficient, cost-effective strategies for intravenous administration. In conclusion, advancing research on the application of curcumin in breast cancer treatment is essential for improving therapeutic outcomes and quality of life for individuals diagnosed with this condition. Additionally, further clinical studies are necessary to establish its role as a reliable breast cancer treatment, ensuring both safety and effectiveness before it can be integrated into standard therapeutic protocols.

## Figures and Tables

**Figure 1 ijms-25-11492-f001:**
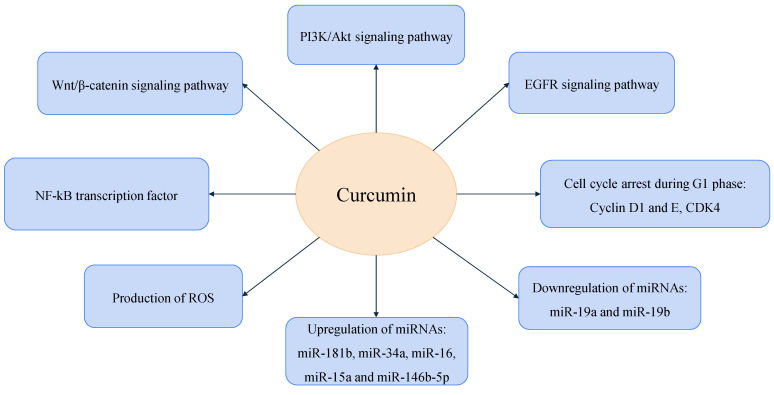
Curcumin engagement in key biological processes to inhibit breast cancer development.

**Figure 2 ijms-25-11492-f002:**
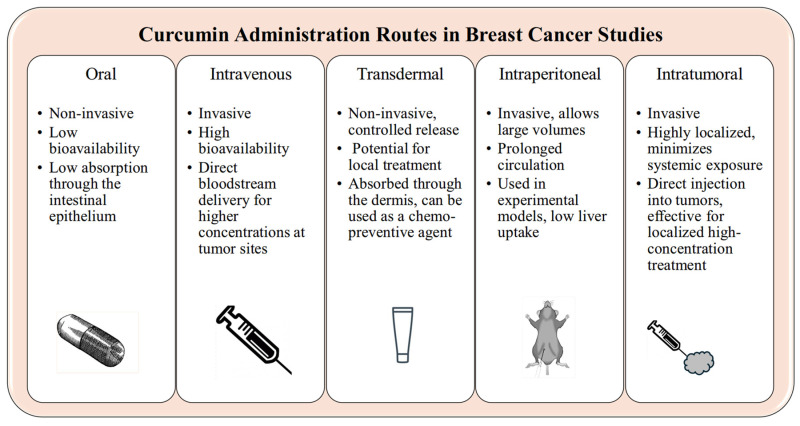
Comparative efficacy of oral, intravenous, transdermal, intraperitoneal, and intratumoral curcumin administration routes in breast cancer therapy.

**Table 2 ijms-25-11492-t002:** Summary of clinical trials assessing the efficacy of curcumin in breast cancer treatment.

Clinical Trial	Study Design	Population	Treatment	Results	Reference
Curcumin + Docetaxel	Phase I	Patients with advanced breast cancer	Curcumin (From 500 mg/day) + Docetaxel (100 mg/m^2^ IV)	Enhanced therapeutic outcomes, safe dose determined	[21]
Curcumin + Radiotherapy (RT)	Randomized, double-blind, placebo-controlled clinical trial	Women with non-inflammatory breast cancer or carcinoma in situ prescribed radiation therapy (RT)	Curcumin (6 g/day)	Reduction in severity of radiation dermatitis	[23]
Curcumin + Polyphenols in Breast Cancer Surgery	Controlled trial with two groups	26 patients with diagnosed breast cancer; and 11 patients in the control group, completed the trial	Curcumin and polyphenols (three capsules/day) from diagnosis to surgery; two capsules 2–6 h before surgery	Metabolites of curcuminoids detected in mammary tissues, suggesting anti-cancer effects with long-term exposure	[82]
Curcumin + Paclitaxel	Randomized, double-blind, placebo-controlled	Patients with advanced and metastatic breast cancer	Curcumin (300 mg) + Paclitaxel (80 mg/m^2^)	Improved objective response rates and reduced side effects	[25]

**Table 3 ijms-25-11492-t003:** Comparison between oral and intravenous routes of curcumin administration in breast cancer treatments.

Criteria	Oral Route Administration	Intravenous Route Administration
Cost	More economical	More expensive
Adherence to treatment	Easier adherence, enhanced tolerability, and suitability for an extended duration	Less adherence, reduced tolerability, and temporary use
Improved formulations	Curcumin into self-nano emulsifying drug delivery system (CPCSNEDDS) [96]. Curcumin solution dissolved in corn oil [38]. D-alpha-tocopheryl polyethylene glycol succinate -coated bilosomes were successfully formulated and loaded with curcumin (TPGS-CUR-Bil) [97]. PLGA-Levan Nanomicelle Loaded with Curcumin 1 mL/day [98]. CUR/IR780@SMEDDS 75 mg/kg (Self-Microemulsifying Drug Delivery System) [99]. Curcumin extracted (Powdered Turmeric rhizomes (500 gm)) with 100% methanol [100]. Curcumin–docetaxel co-loaded nanosuspension 40 mg/kg [101]. Curcumin capsule 500 mg [21]. Capsule of 505 mg of blend (190 mg curcumin extract, 65 mg trans-resveratrol, 125 mg flaxseed extract, and 125 mg red clover extract) [82]. Curcumin (Curcumin C3 Complex^®^) [23].	PEG–albumin–curcumin Nanoparticles 10 mg/mL [102]. Curcumin loaded with an amphiphilic block copolymer (MePEG-peptide-PET-PCL) nanoparticles (Cur-P-NPs) [103]. Cur-FA-CMC/PVP GO NPs [104]. Curcumin-loaded redox-responsive mesoporous silica nanoparticles (MSN/CUR-PEI-FA) [105]. Curcumin-loaded ZnO nanoparticles (ZnO-PBA-Curcumin) 10 mg/kg [106]. PLGA-PEG nanoparticles curcumin delivery system 5 mg/kg [34]. pH/redox nanocarrier CUR@PCPP [49]. Alginate–curcumin conjugate (AA-Cur) [107]. Curcumin nanocrystals (Cur-NC) modified with Hyaluronic Acid (HA@Cur-NC) [108]. Mifepristone and curcumin encapsulated in Hyaluronic Acid-modified liposomes (CUR&RU486/HA-LIPs) [109]. Curcumin-loaded and Hyaluronic Acid-coated ZIF-8 (Cur@ZIF-8@HA) [110]. Curcumin delivered Conjugating Hyaluronic Acid (HA) on the surface of Mesoporous silica nanoparticle (MSN-HA-C) [42]. Curcumin-loaded oxidation-responsive mPEG-b-PLG (Se)-TP polymeric micelle (Cur-loaded micelles) [111]. HA-CHEMS conjugates and TPGS self-assembled into Curcumine-loaded nanoparticles (HA-CHEMS-Cur-TPGS NPs) [112]. Folate-conjugated curcumin and gemcitabine-loaded nanoparticles (FCGNPs) [113]. Methotrexate and Curcumin co-encapsulated in Poly(lactic-co-glycolic acid) (PLGA) nanoparticles (MTX-CUR-NPs) [114]. Heparin-poloxamer P403 (HP403) co-load curcuminoid (Cur) and cisplatin hydrate (CisOH) (HP403@CisOH@Cur) [115]. Co-delivery of paclitaxel and curcumin by biodegradable polymeric nanoparticles (PTX-CUR-NPs) [117]. PEG-PLGA encapsulated paclitaxel and curcumin (PC-NPs) nanoparticles [69]. Curcumin^®^ (CUC-01) 300 mg/week [25].
Combination with other therapies	Gemcitabine [98]. Paclitaxel [100]. Docetaxel [21,101]. IR780 [99]. Radiotherapy [23].	Gemcitabine [113]. Paclitaxel [25,69,117]. Methotrexate [114]. Cisplatin [115]. Doxorubicin [116].

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
