# Peer review of "Curcumin Administration Routes in Breast Cancer Treatment"

_ijms, 2024, doi:10.3390/ijms252111492_

Round 1
Reviewer 1 Report
Comments and Suggestions for Authors
The article entitled "Curcumin Administration Routes in Breast Cancer Treatment" by Bianca Mayo et al. provides valuable insights; however, there are several areas that require attention:
- The authors did not cite clinical trials that support the claims regarding curcumin's efficacy in breast cancer treatment.
- The authors failed to discuss the specific pharmacokinetic properties of curcumin, such as absorption, distribution, metabolism, and excretion, which are key for optimizing dosing regimens and treatment outcomes.
- The authors could address the challenges and limitations of current Curcumin administration in breast cancer treatment, as well as provide future perspectives that may help overcome potential limitations.
- While the authors mentioned that curcumin is poorly water-soluble and has low bioavailability, key factors affecting its efficacy in clinical applications. They did not explore potential solutions to address these challenges in the text.
- The authors should include information on potential side effects, risks, or safety concerns associated with prolonged usage of curcumin.
- The authors should incorporate information on clinical validation or well-designed clinical trials to emphasize the importance of advancing research on curcumin in breast cancer treatment, possibly using separate tables for clarity.
- The authors should distinguish how curcumin treatment could be beneficial and the challenges based on factors such as genetic background, hormonal status, or breast cancer subtypes.
Moderate editing of English language required.
Author Response
Reviewer’s comments #1
- The article entitled "Curcumin Administration Routes in Breast Cancer Treatment" by Bianca Mayo et al. provides valuable insights; however, there are several areas that require attention: The authors did not cite clinical trials that support the claims regarding curcumin's efficacy in breast cancer treatment.
The clinical trials published in peer-reviewed journals are mentioned in the specific sections related to the different route types. Additionally, we have included a new table summarizing clinical trials that demonstrate curcumin's effectiveness in breast cancer treatment (Table 2). Below are examples of the clinical trials included in the manuscript:
Curcumin was administered orally from 500 mg/day and gradually increased for seven consecutive days per cycle, starting four days before and continuing two days after the docetaxel infusion. This study concludes that the established recommended dose for curcumin is 6 g/day for seven consecutive days every three weeks, administered alongside a standard dose of docetaxel [Bayet-Robert, M.; 2010, doi:10.4161/cbt.9.1.10392.]
In a randomized, double-blind, placebo-controlled clinical trial involving 30 breast cancer patients undergoing radiotherapy, oral administration of 6 grams of Curcumin (Curcumin C3 Complex®) daily reduces the severity and incidence of moist desquamation of radiation dermatitis, compared to placebo. These findings suggest that curcumin may offer a promising adjuvant therapy to mitigate radiation-induced skin toxicity in breast cancer patients undergoing radiotherapy [Ryan, J.L.; 2013, doi:10.1667/RR3255.1.].
A subsequent clinical trial involved 37 breast cancer patients, 26 were assigned to the polyphenol group, which received three capsules per day containing curcumin and other polyphenols from diagnosis to surgery (5+2 days), while 11 were in the control group. Additionally, the polyphenol group consumed two capsules between 2–6 hours before surgery. The study detected the presence of metabolites derived from curcuminoids in the mammary tissues of these patients, suggesting that these metabolites might exert anticancer effects with long-term exposure [Ávila‐Gálvez, M.Á.; 2021, doi:10.1002/mnfr.202100163.]
In a clinical trial, a group of women was administered a combination of 300 mg of curcumin and paclitaxel via intravenous route. Upon evaluation of adverse effects, intravenous curcumin exhibited no significant safety concerns nor compromised quality of life of patients. Additionally, the adjunct of curcumin demonstrated a modest yet discernible effect in ameliorating gastrointestinal symptoms and fatigue. Analysis conducted three months post-treatment revealed a notably higher overall response rate in the curcumin group compared to the placebo cohort. This indicates a sustained and superior therapeutic effect of curcumin, extending beyond the cessation of treatment [Saghatelyan, T.; 2020, doi:10.1016/j.phymed.2020.153218]
Furthermore, we acknowledge the relevance of providing robust clinical evidence to support the claims regarding curcumin's efficacy in breast cancer treatment in a new section entitled “Challenges and Future Perspectives”.
2. The authors failed to discuss the specific pharmacokinetic properties of curcumin, such as absorption, distribution, metabolism, and excretion, which are key for optimizing dosing regimens and treatment outcomes.
Thank you for this valuable observation. In response, we expanded on this information in the manuscript, providing a more detailed discussion in section number 3:
“Despite its promising effects in cancer treatment, curcumin faces several challenges that hinder its clinical application. These challenges include its hydrophobic nature, photo-degradability, poor absorption, rapid metabolism and elimination, short half-life, and low bioavailability [Giordano; Tommonaro 2019, doi:10.3390/nu11102376; Dei Cas, 2019, doi:10.3390/nu11092147.]. It is important to note that, when taken orally, curcumin is rapidly metabolized in the liver and intestines through glucuronidation and sulfation at the phenolic site, significantly limiting its absorption and distribution [Dei Cas, M.; 2019, doi:10.3390/nu11092147.]. To address these limitations, various delivery methods aimed at improving curcumin's pharmacokinetics, including specialized formulations such as nanoparticles, liposomes, phospholipid complexes, as well as strategic drug combinations and alternative routes of administration, have shown promising results [Silvestre, F.; 2023, doi:10.3390/ph16070943].”
3. The authors could address the challenges and limitations of current Curcumin administration in breast cancer treatment, as well as provide future perspectives that may help overcome potential limitations.
While the authors mentioned that curcumin is poorly water-soluble and has low bioavailability, key factors affecting its efficacy in clinical applications. They did not explore potential solutions to address these challenges in the text.
The authors should include information on potential side effects, risks, or safety concerns associated with prolonged usage of curcumin.
These comments were expanded upon at the end of the manuscript to ensure a comprehensive discussion of the challenges and future perspectives regarding curcumin administration in breast cancer treatment:
“The therapeutic challenges of using curcumin in breast cancer treatment are primarily related to its low bioavailability, rapid degradation and poor aqueous solubility [Dei Cas, 2019, doi:10.3390/nu11092147; Silvestre, F.; 2023, doi:10.3390/ph16070943.]. Although innovative formulations such as nano-micelles, solid lipid nanoparticles and folic acid-modified liposomes have been developed to enhance its absorption and efficacy [Wang, W.; 2018, doi:10.3390/molecules23071578.; Lin, X.; 2023, doi:10.1016/j.jddst.2022.104050.; Shukla, M.; 2017, doi:10.1080/03639045.2016.1239732; Hegazy, H.; 2022, doi:10.1016/j.ijpharm.2022.121717.; Eskandari, Z.; 2021, doi:10.1016/j.ijbiomac.2021.08.115.; Liu, Y.; 2019, doi:10.2147/IJN.S200847.; Sahu, B.P.; 2016, doi:10.1080/17425247.2016.1182486.; Thadakapally, R.; 2016, doi:10.4103/0250-474X.180250.; Guo, F.; 2019, doi:10.1080/10717544.2019.1676843.; Sahne, F.; 2019, doi:10.1021/acsbiomaterials.8b01628.; Li, N.; 2018, doi:10.1080/21691401.2018.1473412.; Kundu, M.; 2019, doi:10.1016/j.jare.2019.02.036.; Karabasz, A.; 2019, doi:10.2147/IJN.S213942.; Ji, P.; 2020, doi:10.1039/C9BM01605H.; Sun, M.; 2023, doi:10.1016/j.jddst.2023.104956.; Yu, S.; 2021, doi:10.1016/j.colsurfb.2021.111759.; He, H.; 2019, doi:10.1021/acsbiomaterials.9b00212.; Huang, C, 2020, doi:10.2147/IJN.S242490.; Mukhopadhyay, R.; 2020, doi:10.1007/s11095-020-2758-5.; Vakilinezhad, M.A.; 2019, doi:10.1016/j.colsurfb.2019.110515.; Nguyen, N.T.; 2022, doi:10.3390/gels8010059.; Fathy Abd-Ellatef, G.-E.; 2020, doi:10.3390/pharmaceutics12020096.; Xiong, K.; 2020, doi:10.1016/j.ijpharm.2020.119875.], translating these strategies into clinical applications remains a significant challenge. Administering curcumin in a way that maintains its potency while reducing side effects (gastrointestinal disturbances such as nausea, diarrhea, bloating, dermatitis and urticaria) [Sharifi-Rad, 2020, doi: 10.3389/fphar.2020.01021] requires further investigation. Additionally, the variability in patient responses and the need for precise control over curcumin release within the tumor microenvironment are also major challenges.
Looking ahead, overcoming these limitations will require the development of more sophisticated drug delivery systems and personalized therapeutic approaches. Combining curcumin with conventional cancer therapies, such as paclitaxel or doxorubicin, has demonstrated enhanced anticancer effects in preclinical studies, highlighting its potential for synergistic treatment strategies [Saghatelyan, T, 2020, doi:10.1016/j.phymed.2020.153218; Bayet-Robert, M., 2010, doi:10.4161/cbt.9.1.10392]. Further research is also needed to refine delivery methods, which could improve the ability of curcumin to target tumor cells while minimizing off-target effects [Sharifi-Rad, 2020, doi: 10.3389/fphar.2020.01021]. With continued advancements, curcumin could play a more significant role in integrative breast cancer treatment regimens.”
4. The authors should incorporate information on clinical validation or well-designed clinical trials to emphasize the importance of advancing research on curcumin in breast cancer treatment, possibly using separate tables for clarity.
Thank you for your suggestion. We included Table 2 which summarizes key clinical trials involving curcumin in breast cancer treatment.
5. The authors should distinguish how curcumin treatment could be beneficial, and the challenges based on factors such as genetic background, hormonal status, or breast cancer subtypes.
We appreciate the comment. We have now detailed the classification of the breast cancer cell lines used in various studies based on their subtypes. The following lines have been added:
“Numerous studies have employed MCF-7 cells to explore the impacts of different curcumin delivery methods on breast cancer. MCF-7 cell line is ERα+ and PR+, non-invasive and non-tumorigenic in vivo unless supplemented with estrogen [Moon, 2020, doi: 10.1371/journal.pone.0234012] …”
“In vitro studies conducted on MDA-MB 231 cells, a TNBC cell line characterized by elevated expression levels of genes linked to epithelial-mesenchymal transition (EMT), migration and cellular differentiation [Moon, 2020, doi: 10.1371/journal.pone.0234012], showed the effectiveness of a hydrogel formulation containing laponite rapid dispersion (Lap®), chitosan and polyvinyl alcohol, designed specifically for the controlled delivery of curcumin (Lap®CUR/CS@PVA hydrogel) …”
“Similar results were observed when curcumin was loaded onto solid lipid nanoparticles (SLN) in SKBR3 cells, that overexpress HER2, resulting in increased apoptosis [Wang, W.; 2018, doi:10.3390/molecules23071578] …"

Reviewer 2 Report
Comments and Suggestions for Authors
The manuscript titled Curcumin Administration Routes in Breast Cancer Treatment reports the summary of in vitro, in vivo, and clinical studies on the reported anticancer properties of curcumin against breast cancer. This review article is well written and structured. However, the authors did not sufficiently emphasize the molecular heterogeneity of the breast cancer subtypes and their impact on prognosis and clinical outcome. In particular, hormone-dependent breast cancer which has a better prognosis with hormone therapy than triple-negative breast cancer (TNBC), deprived of targeted therapies, is a therapeutically challenging subtype. In addition, for breast cancer cell lines, the authors should clearly state whether these cells are hormone receptor-positive, poorly invasive (i.e., MCF-7), or non-hormone-dependent and highly invasive (i.e., MDA-MB-231). Of course, we are more interested in the anticancer properties of curcumin against TNBC cells than MCF-7 cells, whose targeted endocrine therapies are very effective in hormone-dependent breast cancer patients. There is one minor comment; key acronyms/abbreviations must be defined such as HER2, CDK, NLP, and ALDH.
Comments on the Quality of English LanguageMinor editing of English language is required.
Author Response
Reviewer’s comments #2
- The manuscript titled Curcumin Administration Routes in Breast Cancer Treatment reports the summary of in vitro, in vivo, and clinical studies on the reported anticancer properties of curcumin against breast cancer. This review article is well written and structured. However, the authors did not sufficiently emphasize the molecular heterogeneity of the breast cancer subtypes and their impact on prognosis and clinical outcome. Hormone-dependent breast cancer which has a better prognosis with hormone therapy than triple-negative breast cancer (TNBC), deprived of targeted therapies, is a therapeutically challenging subtype.
Thank you for your suggestion. We have added the following paragraph to the introduction.
“Breast cancer is classified into different subtypes based on the presence or absence of specific protein markers, namely estrogen receptor α (ERα), progesterone receptor (PR) and ERBB2 (also known as Human Epidermal Growth Factor Receptor 2, HER2). Hormone-dependent subtypes, such as ER+ and PR+ breast cancers, typically have a more favorable prognosis, as they respond well to hormone therapies like tamoxifen, which target the hormone signaling pathways that drive cancer growth. HER2+ breast cancers, on the other hand, respond to monoclonal antibody treatments such as pertuzumab, which binds to an extracellular domain of the HER2 receptor. In contrast, triple-negative breast cancer (TNBC), characterized by the absence of ERα, PR and ERBB2 expression, is considered more aggressive and presents significant therapeutic challenges, often resulting in poorer clinical outcomes due to the lack of targeted therapies [Petri, B.J, 2020, doi:10.1007/s10555-020-09905-7.;Cosar, 2022, doi: 10.2147/BCTT.S380754]. For that reason, looking for novel alternative treatments for breast cancer is a must to reduce patient suffering and death rates.”
- For breast cancer cell lines, the authors should clearly state whether these cells are hormone receptor-positive, poorly invasive (i.e., MCF-7), or non-hormone-dependent and highly invasive (i.e., MDA-MB-231). Of course, we are more interested in the anticancer properties of curcumin against TNBC cells than MCF-7 cells, whose targeted endocrine therapies are very effective in hormone-dependent breast cancer patients.
Thank for this comment we added the next lines:
“Numerous studies have employed MCF-7 cells to explore the impacts of different curcumin delivery methods on breast cancer. MCF-7 cell line is ERα+ and PR+, non-invasive and non-tumorigenic in vivo unless supplemented with estrogen [Moon, 2020, doi: 10.1371/journal.pone.0234012] …”
“In vitro studies conducted on MDA-MB 231 cells, a TNBC cell line characterized by elevated expression levels of genes linked to epithelial-mesenchymal transition (EMT), migration and cellular differentiation [Moon, 2020, doi: 10.1371/journal.pone.0234012], showed the effectiveness of a hydrogel formulation containing laponite rapid dispersion (Lap®), chitosan and polyvinyl alcohol, designed specifically for the controlled delivery of curcumin (Lap®CUR/CS@PVA hydrogel) …”
“Similar results were observed when curcumin was loaded onto solid lipid nanoparticles (SLN) in SKBR3 cells, that overexpress HER2, resulting in increased apoptosis [Wang, W.; 2018, doi:10.3390/molecules23071578] …”
- There is one minor comment; key acronyms/abbreviations must be defined such as HER2, CDK, NLP, and ALDH.
Thank you for your observation. We ensure that acronyms and abbreviations are clearly defined upon their first mention in the manuscript to improve clarity for readers.

Reviewer 3 Report
Comments and Suggestions for Authors
This review paper summarizes the administration routes of curcumin in breast cancer treatment, comparing the effect between oral and intravenous routes. Some previous papers reviewed the effect of curcumin on all cancers, while this review paper focuses on breast cancer. However, this paper, especially the introduction section, did not describe the specific characteristics of breast cancer in terms of curcumin treatment. More importantly, most studies summarized in this paper were conducted using cell lines and mice models. This was consistent with the current guidelines from the US NIH, that the evidence is currently inadequate to recommend curcumin-containing products for the treatment of cancer. However, this lack of evidence was not mentioned in this review paper. This paper describes curcumin as a promising agent for breast cancer treatment. Therefore, I suggest rejection of this review paper.
Author Response
Reviewer’s comment #3
The introduction section did not describe the specific characteristics of breast cancer in terms of curcumin treatment. More importantly, most studies summarized in this paper were conducted using cell lines and mice models. This was consistent with the current guidelines from the US NIH, that the evidence is currently inadequate to recommend curcumin-containing products for the treatment of cancer. However, this lack of evidence was not mentioned in this review paper. Therefore, I suggest rejection of this review paper.
Thank you for your sincere response. We have added more information about breast cancer subtypes in the introduction, and how novel therapies are necessary in aggressive cancer types:
“Breast cancer is classified into different subtypes based on the presence or absence of specific protein markers, namely estrogen receptor α (ERα), progesterone receptor (PR) and ERBB2 (also known as Human Epidermal Growth Factor Receptor 2, HER2). Hormone-dependent subtypes, such as ER+ and PR+ breast cancers, typically have a more favorable prognosis, as they respond well to hormone therapies like tamoxifen, which target the hormone signaling pathways that drive cancer growth. HER2+ breast cancers, on the other hand, respond to monoclonal antibody treatments such as pertuzumab, which binds to an extracellular domain of the HER2 receptor. In contrast, triple-negative breast cancer (TNBC), characterized by the absence of ERα, PR and ERBB2 expression, is considered more aggressive and presents significant therapeutic challenges, often resulting in poorer clinical outcomes due to the lack of targeted therapies [Petri, B.J, 2020, doi:10.1007/s10555-020-09905-7.;Cosar, 2022, doi: 10.2147/BCTT.S380754]. For that reason, looking for novel alternative treatments for breast cancer is a must to reduce patient suffering and death rates.”
The clinical trials published in peer-reviewed journals are mentioned in the specific sections related to the different route types. We have also included a new table of clinical trials demonstrating the effectiveness of curcumin in breast cancer treatment (Table 2).
Below are examples of the clinical trials included in the manuscript:
Curcumin was administered orally from 500 mg/day and gradually increased for seven consecutive days per cycle, starting four days before and continuing two days after the docetaxel infusion. This study concludes that the established recommended dose for curcumin is 6 g/day for seven consecutive days every three weeks, administered alongside a standard dose of docetaxel [Bayet-Robert, M.; 2010, doi:10.4161/cbt.9.1.10392.]
In a randomized, double-blind, placebo-controlled clinical trial involving 30 breast cancer patients undergoing radiotherapy, oral administration of 6 grams of Curcumin (Curcumin C3 Complex®) daily reduces the severity and incidence of moist desquamation of radiation dermatitis, compared to placebo. These findings suggest that curcumin may offer a promising adjuvant therapy to mitigate radiation-induced skin toxicity in breast cancer patients undergoing radiotherapy [Ryan, J.L.; 2013, doi:10.1667/RR3255.1.].
A subsequent clinical trial involved 37 breast cancer patients, 26 were assigned to the polyphenol group, which received three capsules per day containing curcumin and other polyphenols from diagnosis to surgery (5+2 days), while 11 were in the control group. Additionally, the polyphenol group consumed two capsules between 2–6 hours before surgery. The study detected the presence of metabolites derived from curcuminoids in the mammary tissues of these patients, suggesting that these metabolites might exert anticancer effects with long-term exposure [Ávila‐Gálvez, M.Á.; 2021, doi:10.1002/mnfr.202100163.]
In a clinical trial, a group of women was administered a combination of 300 mg of curcumin and paclitaxel via intravenous route. Upon evaluation of adverse effects, intravenous curcumin exhibited no significant safety concerns nor compromised quality of life of patients. Additionally, the adjunct of curcumin demonstrated a modest yet discernible effect in ameliorating gastrointestinal symptoms and fatigue. Analysis conducted three months post-treatment revealed a notably higher overall response rate in the curcumin group compared to the placebo cohort. This indicates a sustained and superior therapeutic effect of curcumin, extending beyond the cessation of treatment [Saghatelyan, T.; 2020, doi:10.1016/j.phymed.2020.153218]
Finally, we acknowledge the relevance of providing robust clinical evidence to support the claims regarding curcumin's efficacy in breast cancer treatment in a new section entitled “Challenges and Future Perspectives”.
“The therapeutic challenges of using curcumin in breast cancer treatment are primarily related to its low bioavailability, rapid degradation and poor aqueous solubility [Dei Cas, 2019, doi:10.3390/nu11092147; Silvestre, F.; 2023, doi:10.3390/ph16070943.]. Although innovative formulations such as nano-micelles, solid lipid nanoparticles and folic acid-modified liposomes have been developed to enhance its absorption and efficacy [Wang, W.; 2018, doi:10.3390/molecules23071578.; Lin, X.; 2023, doi:10.1016/j.jddst.2022.104050.; Shukla, M.; 2017, doi:10.1080/03639045.2016.1239732; Hegazy, H.; 2022, doi:10.1016/j.ijpharm.2022.121717.; Eskandari, Z.; 2021, doi:10.1016/j.ijbiomac.2021.08.115.; Liu, Y.; 2019, doi:10.2147/IJN.S200847.; Sahu, B.P.; 2016, doi:10.1080/17425247.2016.1182486.; Thadakapally, R.; 2016, doi:10.4103/0250-474X.180250.; Guo, F.; 2019, doi:10.1080/10717544.2019.1676843.; Sahne, F.; 2019, doi:10.1021/acsbiomaterials.8b01628.; Li, N.; 2018, doi:10.1080/21691401.2018.1473412.; Kundu, M.; 2019, doi:10.1016/j.jare.2019.02.036.; Karabasz, A.; 2019, doi:10.2147/IJN.S213942.; Ji, P.; 2020, doi:10.1039/C9BM01605H.; Sun, M.; 2023, doi:10.1016/j.jddst.2023.104956.; Yu, S.; 2021, doi:10.1016/j.colsurfb.2021.111759.; He, H.; 2019, doi:10.1021/acsbiomaterials.9b00212.; Huang, C, 2020, doi:10.2147/IJN.S242490.; Mukhopadhyay, R.; 2020, doi:10.1007/s11095-020-2758-5.; Vakilinezhad, M.A.; 2019, doi:10.1016/j.colsurfb.2019.110515.; Nguyen, N.T.; 2022, doi:10.3390/gels8010059.; Fathy Abd-Ellatef, G.-E.; 2020, doi:10.3390/pharmaceutics12020096.; Xiong, K.; 2020, doi:10.1016/j.ijpharm.2020.119875.], translating these strategies into clinical applications remains a significant challenge. Administering curcumin in a way that maintains its potency while reducing side effects (gastrointestinal disturbances such as nausea, diarrhea, bloating, dermatitis and urticaria) [Sharifi-Rad, 2020, doi: 10.3389/fphar.2020.01021] requires further investigation. Additionally, the variability in patient responses and the need for precise control over curcumin release within the tumor microenvironment are also major challenges.
Looking ahead, overcoming these limitations will require the development of more sophisticated drug delivery systems and personalized therapeutic approaches. Combining curcumin with conventional cancer therapies, such as paclitaxel or doxorubicin, has demonstrated enhanced anticancer effects in preclinical studies, highlighting its potential for synergistic treatment strategies [Saghatelyan, T, 2020, doi:10.1016/j.phymed.2020.153218; Bayet-Robert, M., 2010, doi:10.4161/cbt.9.1.10392]. Further research is also needed to refine delivery methods, which could improve’s the ability of curcumin to target tumor cells while minimizing off-target effects [Sharifi-Rad, 2020, doi: 10.3389/fphar.2020.01021]. With continued advancements, curcumin could play a more significant role in integrative breast cancer treatment regimens.”

Round 2
Reviewer 1 Report
Comments and Suggestions for Authors
Accept in present form
Comments on the Quality of English LanguageModerate editing of English language required.
Author Response
We greatly appreciate your insightful comments and suggestions, which have undoubtedly enhanced our work.
We carefully considered all the suggestions and comments and have made the necessary revisions accordingly.
Reviewer 2 Report
Comments and Suggestions for Authors
I would like to thank the authors for considering my comments, which improved the quality of the manuscript.
Comments on the Quality of English LanguageMinor editing English language is still required.
Author Response

(The authors gave the same response as above.)

Reviewer 3 Report
Comments and Suggestions for Authors
My concerns were not addressed in the revised manuscript.
(1) In the revised manuscript, only a description about breast cancer subtypes was added in the first paragraph of the introduction. From line 43 to line 78, there are only general descriptions about curcumin on all cancers. Then line 80 claimed that "In this sense, curcumin has emerged as a promising candidate for breast cancer treatment". The conclusion for promising candidate was not supported based on the introduction.
(2) Although the added Table 2 described some clinical trials, it does not discuss whether these evidence are strong to support curcumin for treatment. Also, the guidelines from the US NIH was even not mentioned or discussed in the manuscipt.
Author Response
We greatly appreciate your insightful comments and suggestions, which have undoubtedly enhanced our work. We carefully considered all the suggestions and comments and have made the necessary revisions accordingly.
Below, we point out each observation provided (in blue) and its corresponding response (in black):
Reviewer’s comments #3
My concerns were not addressed in the revised manuscript.
(1) In the revised manuscript, only a description about breast cancer subtypes was added in the first paragraph of the introduction. From line 43 to line 78, there are only general descriptions about curcumin on all cancers. Then line 80 claimed that "In this sense, curcumin has emerged as a promising candidate for breast cancer treatment". The conclusion for promising candidate was not supported based on the introduction.
Thank you for giving detailed suggestions that enhance the quality of our manuscript. We added a paragraph in the introduction section summarizing the evidence supporting curcumin as a potential treatment for breast cancer. We tried to give a cohesive introduction for a better comprehension of the main aspects related to breast cancer and curcumin.
“In the context of breast cancer, curcumin has been demonstrated effective in ER+ and TNBC [Guneydas, 2022, DOI:10.31557/APJCP.2022.23.3.911]. In vitro, curcumin reduces proliferation, induces apoptosis and inhibits the spread of breast cancer cells. In vivo, curcumin reduces tumor growth in experimental breast cancer models [Barcelos, 2022, https://doi.org/10.3390/cancers14092165]. In clinical trials, curcumin prevents breast cancer progression and decreases tumor markers when is administrated with docetaxel [Bayet-Robert, 2010, doi:10.4161/cbt.9.1.10392]. Moreover, curcumin reduces conventional cancer therapy side effects such as radiation dermatitis [Ryan, J.L.; 2013, doi:10.1667/RR3255.1.][ Rao S, 2017, 10.3390/medicines4030043].”
(2) Although the added Table 2 described some clinical trials, it does not discuss whether these evidence are strong to support curcumin for treatment. Also, the guidelines from the US NIH was even not mentioned or discussed in the manuscipt.
We have added information from the NCI NIH (U.S.) about curcumin in cancer treatment in the Challenges and Future Perspective section. We also mention that the evidence from clinical trials needs to be confirmed by more extensive clinical trials.
“Finally, the U.S. National Cancer Institute has reported on clinical trials investigating the use of curcumin in cancer prevention and treatment, as well as its role as an adjuvant in conventional therapies, demonstrating its safety and effectiveness [Curcumin (Curcuma, Turmeric) and Cancer (PDQ®)–Health Professional Version was originally published by the National Cancer Institute.]. However, these results must be confirmed through more extensive trials involving larger sample sizes before curcumin can be recommended as a breast cancer treatment.”
Round 3
Reviewer 3 Report
Comments and Suggestions for Authors
In the modified manuscript, the NIH current guideline was not described clearly and the manuscript can be misleading. The NIH guideline emphasizes the insufficient evidence for curcumin for cancer treatment. In the modified manuscript, it described like the NIH reported on clinical trials to demonstrate its safety and effectiveness. This should be corrected.
Author Response
Reviewer’s comments #3
In the modified manuscript, the NIH current guideline was not described clearly, and the manuscript can be misleading. The NIH guideline emphasizes the insufficient evidence for curcumin for cancer treatment. In the modified manuscript, it described like the NIH reported on clinical trials to demonstrate its safety and effectiveness. This should be corrected.
Response: Thank you for your insightful comment. In response, we have revised the manuscript and changed this paragraph
“Finally, it is essential to note that, according to the U.S. National Cancer Institute, while curcumin has been investigated for its potential role in cancer treatment, findings from early-stage trials require confirmation in larger, well-powered studies assessing its safety and effectiveness. The NIH emphasizes that, at present, there is insufficient evidence to recommend curcumin-containing products for cancer prevention or treatment [120]. Further research is necessary to conclusively establish its safety and therapeutic efficacy"
We also added the next sentence in the conclusion section:
“Additionally, further clinical studies are necessary to establish its role as a reliable breast cancer treatment, ensuring both safety and effectiveness before it can be integrated into standard therapeutic protocols.”
Round 4
Reviewer 3 Report
Comments and Suggestions for Authors
All my concerns have been addressed.